# Estimating affective polarization on a social network

Marilena Hohmann[1]*, Michele Coscia[2]

1 Copenhagen Center for Social Data Science, University of Copenhagen, Copenhagen, Denmark, 2 CS Department, IT University of Copenhagen, Copenhagen, Denmark

* marilena.hohmann@sodas.ku.dk

## Abstract

Concerns about polarization and hate speech on social media are widespread. Affective polarization, i.e., hostility among partisans, is crucial in this regard as it links political disagreements to hostile language online. However, only a few methods are available to measure how affectively polarized an online debate is, and the existing approaches do not investigate jointly two defining features of affective polarization: hostility and social distance. To address this methodological gap, we propose a network-based measure of affective polarization that combines both aspects – which allows them to be studied independently. We show that our measure accurately captures the relation between the level of disagreement and the hostility expressed towards others (affective component) and whom individuals choose to interact with or avoid (social distance component). Applying our measure to a large-scale Twitter data set on COVID-19, we find that affective polarization was low in February 2020 and increased to high levels as more users joined the Twitter discussion in the following months.

## Introduction

Affective polarization describes the affective attitude towards like-minded and disagreeing others. Traditionally, surveys are used to measure affective polarization [1]. Survey respondents are, for instance, asked to indicate how they feel toward others with opposing political views or whether they would want an out-group member to be their colleague or friend. When participants feel more hostile towards the out-group than the in-group, affective polarization is high.

Social identity and group dynamics are central to affective polarization, but collecting information on social ties in surveys is challenging. Since this information is usually unavailable in survey data, a relevant conceptual component of affective polarization often remains undiscovered in these standard survey measures. Moreover, while surveys can provide detailed information on attitudes and intentions, this data might not necessarily reflect actual behavior [2]. Although a respondent might claim to dislike or avoid others they disagree with, a survey cannot conclusively clarify whether the participant indeed acts upon these intentions in everyday life [1,3].

Complementary to surveys, social media has emerged as a relevant domain for studying affective polarization [4–6]. On social media, large-scale group dynamics can be

the results of the synthetic experiments is provided in the paper.

**Funding:** The author(s) received no specific funding for this work.

**Competing interests:** The authors have declared that no competing interests exist.

directly observed by considering the social network structures through which users interact. This raises the question: How can we measure affective polarization in a social network?

The most common approach to estimating affective polarization examines the average feelings or sentiment toward the in-group versus out-group [4,6–10]. This measure requires survey respondents or social media users to be divided into two groups. The resulting quantification of affective polarization is relatively crude because people are split into two groups regardless of nuances in political attitudes, such as being moderately conservative or extremely liberal.

A second type of approach involves correlational methods to quantify, for instance, the relation between the strength of policy preferences and how individuals feel toward others with different political ideas [11]. An advantage of this approach is that it is more nuanced than splitting the respondents into two categories. However, neither of the approaches discussed so far can account for how social interactions are organized, which is a crucial indicator of affective polarization.

Some network-based methods explicitly modeling those social relations exist, such as a measure proposed in [5]. The authors split Twitter users into two groups and then measure the sentiment of the interactions within and between groups. While this method considers the social network, it still requires users to be split into two camps. As argued above, such a measure cannot account for fine-grained distinctions between political leanings.

Another group of network-based methods analyzes signed graphs [12–14]. In a signed network, each edge has a positive or negative sign, depending on the type of interaction between two individuals. Measures on signed graphs can capture how the positive or negative interactions align with the network structure. However, they cannot account for nuances in how people interact, such as mild vs. extreme hostility, since the edge signs are too coarse to capture this information.

In summary, current measures of affective polarization face three main challenges: they are insensitive to information on individuals' opinions, the valence of their interactions, or the structure of these interactions. In this paper, we propose a measure that overcomes these issues. Firstly, our method explicitly considers the structure of interactions in a social network. This allows us to quantify social distance, i.e., whether individuals avoid others they disagree with. Secondly, the proposed measure captures the relationship between political opinions and animosity. Our framework models both opinions and hostility as continuous variables. Consequently, we do not need to divide individuals into two groups or compress information regarding the hostility between people into binary edge signs.

In experiments on synthetic data, we show how our measure summarizes the relation between disagreement and hostility (affective component) and information on social interactions (social distance component) in a single affective polarization score. Since the components are distinct, the method allows for the estimation of affective polarization even without considering the social distance component, as in some frameworks the social distance component is regarded as a different dimension, rather than a component, of affective polarization [15].

Moreover, we compare our score to the alternative methods we review above. Subsequently, we apply this measure to a data set of approximately 47 million tweets discussing the COVID-19 pandemic, which US-based users posted to Twitter between February 2020 and July 2020. We find that the Twitter debate was not affectively polarized in February 2020 but became increasingly polarized as more users joined the discussion in the subsequent months.

## Affective polarization measure

The survey-based literature has developed various questionnaire items to measure affective polarization, which can be grouped into two categories: items focusing on hostility and items focusing on social distance. Following this literature, we define two components of affective polarization:

- **Affective component:** Since affective polarization describes increasing dislike or distrust between political opponents, affect is a central element to consider [1,2,9,11,16–20]. While affective polarization can refer to either in-group favoritism or out-group hostility, empirical studies have often prioritized out-group hostility [1,16,21]. Therefore, we focus on hostility here and, in particular, on the relationship between disagreement and hostility. If there is no relation, each individual treats all others in an equally (non-)hostile way, regardless of their opinions, and affective polarization is therefore low (Fig 1a, example on the left). However, if people are increasingly hostile the more they disagree, then affective polarization is high (Fig 1a, example on the right).
- **Social distance component:** Prior survey research has used social distance as an indicator of affective polarization [2,9,11,22]. Social distance survey items assess people's willingness to form social ties across political divides, such as having an out-group member as a friend, neighbor, or in-law [2]. These studies demonstrate that affective polarization shapes social interactions by influencing whom individuals choose to engage with or avoid.

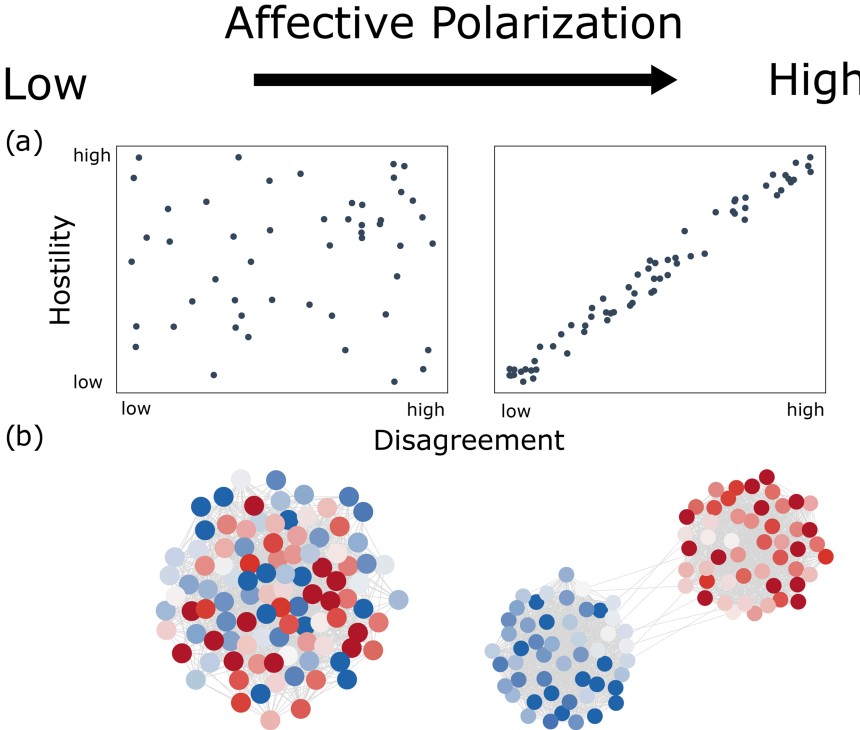

**Fig 1. Components of affective polarization.** (a) Affective component: no relation between disagreement and hostility on the left, strong correlation between disagreement and hostility on the right. (b) Social distance component: individuals are randomly connected in the left graph, whereas there are distinguishable, politically aligned communities in the right graph.

This aspect of affective polarization can be observed by analyzing the structure of interactions between individuals in a social network: If people interact regardless of their political opinion, social distance is small (Fig 1b, example on the left). In contrast, if people interact with like-minded individuals but avoid communities holding opposing views, social distance is large. In social media research, this aspect is also referred to as interactional polarization [6,7].

## Definition

Since both affect and social distance describe the same underlying concept of affective polarization, we argue that a comprehensive measure should capture both components in a single score. We define such a measure in the following section.

## Formulation

We consider a connected, undirected, and unweighted network $G = (V, E)$ with $V$ as the set of individuals and $E \subseteq V \times V$ as the set of connections. We record an opinion value $o_i \in [-1, 1]$ per node in $V$. For each edge between two nodes $i$ and $j$, we determine a disagreement value $x_{i,j}$ as the absolute opinion difference $x_{i,j} = |o_i - o_j|$ for each pair of nodes directly connected by an edge. Moreover, we document a hostility value $y_{i,j} \in [0, 1]$ per edge. An edge represents an interaction between two individuals, such as a reply or an @-mention on a social media platform or a friendship tie in an offline context. We assume that the edges in $E$ are undirected because we want to examine how often and how hostile the interactions between individuals with different political views are. In this macro-level view, it is irrelevant whether liberals address conservatives in a hostile manner or vice versa.

We represent the $o_i$, $x_{i,j}$, and $y_{i,j}$ values as vectors. Given the network $G$, opinions $o$, and hostility $y$, our measure $\alpha_{o,y,G}$ quantifies the relation between disagreement and hostility (affective component) while considering the structure of social interactions (social distance component). To outline how $\alpha_{o,y,G}$ is defined, we first discuss how the two components can be measured separately.

**Affective component.** To quantify the affective component, we use the Spearman rank correlation coefficient $\tilde{\rho}_{o,y}$, which measures the monotonic relationship between two variables. It is well-suited to capture the association between the disagreement vector $x$ and the hostility vector $y$, regardless of whether this relationship is linear. As our Twitter analysis shows, real-world data may be skewed and exhibit nonlinear relationships (see Sect 3 of the S1 File). We therefore opt for the Spearman rank correlation coefficient in the definition of our measure.

Various scenarios are possible: If $\tilde{\rho}_{o,y} > 0$, increasing disagreement coincides with increasing hostility. In other words, there is polarization in the affective component. If $\tilde{\rho}_{o,y} = 0$, disagreement and hostility are unrelated, and there is no affective polarization. If $\tilde{\rho}_{o,y} < 0$, we do not observe affective polarization. In this case, disagreement and hostility are negatively related; i.e., the more people agree, the more hostile they are toward each other. Although it is rather unlikely that this will occur empirically, it is relevant for our measure to distinguish this scenario from the others. The Spearman rank correlation coefficient is defined as follows:

$$\tilde{\rho}_{o,y} = 1 - \frac{6 \sum d_{i,j}^2}{n(n^2 - 1)}$$

where $d_{i,j}$ is the difference in rank between the disagreement values $x_{i,j} = |o_i - o_j|$ and the hostility values $y_{i,j}$ for each connected node pair in $G$, and $n$ is the number of such pairs.

As argued above, we use the Spearman correlation because real-world data may be skewed. However, the Pearson correlation $\rho_{o,y}$ could also be used if the data exhibits a linear relationship. A definition of the Pearson correlation coefficient is provided in Sect 1 of the S1 File.

**Social distance component.** The social distance component describes the structure of interactions between people with similar vs. opposing political opinions. We rely on the recently introduced Generalized Euclidean (GE) polarization measure $\delta_{G,o}$ [23] to quantify this component. By combining opinion data with network information, $\delta_{G,o}$ can quantify 1) the distribution of opinions, 2) the level of structural separation in the network, and 3) how the opinions are aligned with the network structure. The measure is defined as [23]:

$$\delta_{G,o} = \sqrt{(o^+ - o^-)^T L^\dagger (o^+ - o^-)}$$

where $o^+$ is a vector containing all positive opinions and zero otherwise, $o^-$ is a vector containing the absolute value of all negative opinions and zero otherwise, and $L^\dagger$ is the pseudoinverse of the Laplacian matrix of the network $G$.

**Affective polarization measure $\alpha_{o,y,G}$.** By combining the Pearson correlation and the Generalized Euclidean distance, we can obtain an affective polarization score that includes both the affective and the social distance component:

$$\alpha_{o,y,G} = \tilde{\rho}_{o,y}\delta_{G,o}.$$

We multiply the two components to preserve the sign flip from a negative to a positive correlation in $\tilde{\rho}_{o,y}$. The sign of our measure $\alpha_{o,y,G}$ can therefore be interpreted similarly to the sign of the Spearman rank correlation: if $\alpha_{o,y,G} > 0$, there is affective polarization; if $\alpha_{o,y,G} = 0$, there is no affective polarization; and if $\alpha_{o,y,G} < 0$, there is a third scenario in which low disagreement co-occurs with high hostility.

In general, $\alpha_{o,y,G}$ takes values from an arbitrary negative to an arbitrary positive number. The higher the $\alpha_{o,y,G}$ value, the higher the level of affective polarization.

## Results

We conduct experiments with synthetic data to demonstrate how our measure captures both the affective component and the social distance component and to compare it to other methods currently used in the literature. Subsequently, we apply our measure to a series of real-world Twitter networks in which users discuss COVID-19 restrictions.

### Method validation

We compare our measure $\alpha_{o,y,G}$ to other affective polarization methods proposed in the literature:

- $\mu_{o,y}$: The average sentiment score $\mu_{o,y}$ enables a comparison between the mean in-group vs. mean out-group hostility [4,6–10]. We report two values: *Avg $\mu_{o,y}$ (Like-minded)* which summarizes the average hostility for all like-minded node pairs, and *Avg $\mu_{o,y}$ (Cross-cutting)* which summarizes the average hostility among all disagreeing node pairs.
- $\rho_{o,y}$: The Pearson correlation coefficient $\rho_{o,y}$ measures the linear relationship between the disagreement vector $x$ and hostility vector $y$.

- *$EMD_{o,y,G}$*: This measure, which combines the Earth Mover's Distance and Krackhardt's E/I index, focuses on network-based interactions [5]. It splits users into two groups and measures sentiment within and between these groups. For each group, we calculate one score that captures the difference between in-group versus out-group hostility.
- *$SAI_{o,y,G}$*: The Structural Alignment Index $SAI_{o,y,G}$ examines the alignment between edge signs and network structure by measuring frustration in a signed network [12].
- *$POLE_{y,G}$*: This method calculates the node-level Pearson correlation between transition probabilities on a signed and unsigned network. The final polarization measure, $POLE_{y,G}$, is the average of these node-level scores [13].

Sect 1 in the S1 File contains further details of how these measures are defined. Some of the alternative measures, $\mu_{o,y}$, $EMD_{o,y,G}$, and $SAI_{o,y,G}$, require the nodes to be split into two groups. These groups could, for instance, represent climate change believers and disbelievers [5] or Democrats and Republicans. In the experiments below, we split the nodes into a blue and a red group. All values shown in the following figures represent the lower and upper bound of a 95% confidence interval, which we calculate across 100 experiment repetitions.

**The affective component.**

The affective component describes the relationship between disagreement and hostility. Since this relation is characterized by the joint distribution of disagreement values $x$ and hostility values $y$, a comprehensive measure should quantify both aspects. Therefore, we split the affective component experiments into two subsections. First, we test the measures' sensitivity to changes in the hostility distribution, followed by changes in the disagreement distribution.

**Hostility Distribution.** Fig 2 summarizes the experiment on changes in the hostility distribution. We generate a random graph and assign opinion values from −1 to +1 to the nodes. The network structure and the opinion values stay fixed for all experiments (a) to (e). As shown in the scatter plots in Fig 2, we only change the hostility value assigned to each edge between two nodes.

In (a) and (b), hostility decreases as disagreement increases. This relation gets weaker from (a) to (b); and in (c), the two variables are entirely uncorrelated. In (d) and (e), hostility and disagreement are positively related, and we expect the highest level of affective polarization in these cases.

The affective polarization measure $\alpha_{o,y,G}$, the Pearson correlation $\rho_{o,y}$, and $EMD_{o,y,G}$ confirm this expectation as they increase from (a) to (e). We conclude that these measures can account for changes in the hostility distribution.

Similarly, the average sentiment score $\mu_{o,y}$ can, for the most part, account for changes in the hostility distribution. As expected, the average hostility among like-minded individuals *Avg $\mu_{o,y}$ (Like-minded)* decreases from (a) to (e). However, *Avg $\mu_{o,y}$ (Cross-cutting)* cannot consistently capture the increasing hostility among disagreeing individuals as the confidence intervals overlap.

Lastly, the measures working with signed networks, $SAI_{o,y,G}$ and $POLE_{y,G}$ cannot fully capture changes in the hostility distribution. While $SAI_{o,y,G}$ accounts for the sign flip going from a negative disagreement–hostility relation to a positive one, the measure is not sensitive to the strength of the correlation. The values in (a) and (b) are the same, and so are the values in (d) and (e). The same shortcoming applies to $POLE_{y,G}$. Moreover, $POLE_{y,G}$ cannot capture the sign of the correlation changing, and it returns the highest values in example (c), where disagreement and hostility are entirely uncorrelated.

**Disagreement distribution.** We now turn to the second aspect of the affective component: the distribution of disagreement values. As before, we generate a random graph and assign opinion values between −1 and +1 to the nodes. We update the opinions of a few nodes

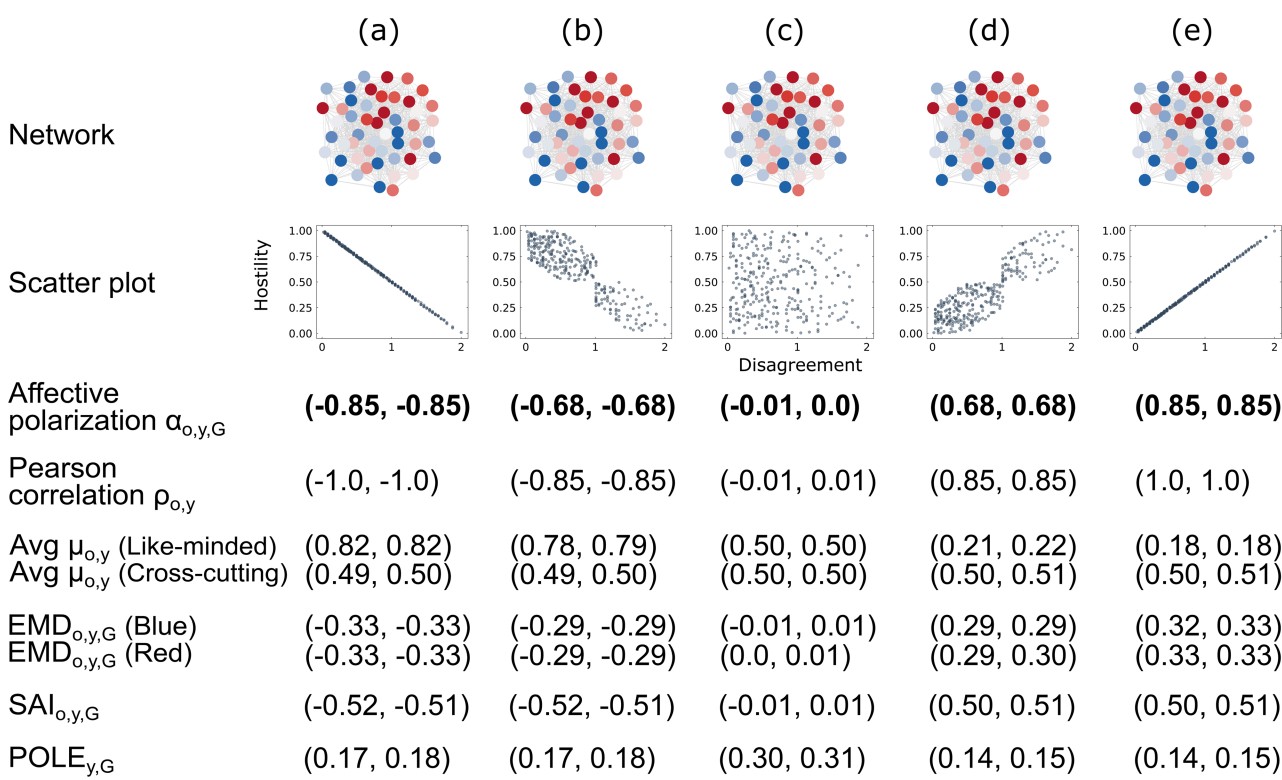

**Fig 2. Affective component (hostility distribution).** From top to bottom: network where the node color reflects opinion, from −1 (blue), passing 0 (white), to +1 (red); scatter plot with disagreement on the x-axis and hostility on the y-axis; values of $\alpha_{o,y,G}$, $\rho_{o,y}$, $\mu_{o,y}$, $EMD_{o,y,G}$, $SAI_{o,y,G}$, and $POLE_{y,G}$. The results (in parentheses) denote the lower and upper bound of the 95% confidence interval across 100 repetitions of the experiment.

so that the overall level of disagreement increases slightly from (a) to (e). Then, we assign hostility values to the edges. Importantly, we use the same hostility vector $y$ across all the examples in Fig 3. Since the disagreement values $x$ change, while the hostility values $y$ stay fixed, the correlation between $x$ and $y$ gets stronger, as shown in the scatter plots in Fig 3.

We find that both the affective polarization measure $\alpha_{o,y,G}$ and the Pearson correlation $\rho_{o,y}$ are sensitive to changes in the disagreement distribution. Conversely, $\mu_{o,y}$, $EMD_{o,y,G}$, $SAI_{o,y,G}$, and $POLE_{y,G}$ cannot distinguish between any of the examples (a) to (e). These measures split the nodes into groups and, therefore, cannot capture changes in the in-group disagreement distribution. Importantly, these differences cause the disagreement–hostility relation, and thus the level of affective polarization, to increase from (a) to (e). In short, $\mu_{o,y}$, $EMD_{o,y,G}$, $SAI_{o,y,G}$, and $POLE_{y,G}$ are insensitive to this aspect of the affective component.

**The social distance component.** The third experiment focuses on the social distance component. We generate a network with a community of blue nodes and a community of red nodes. The opinion, disagreement, and hostility values are fixed across the examples (a) to (e). Consequently, the disagreement–hostility relation is the same for all networks, as shown in the scatter plots in Fig 4. However, the network structure changes from (a) to (e). As the connections between the two communities become sparser, social distance increases, and we therefore expect the affective polarization level to increase from (a) to (e).

Fig 4 confirms that our affective polarization measure $\alpha_{o,y,G}$ is sensitive to the social distance component: $\alpha_{o,y,G}$ increases from (a) to (e) as expected. $POLE_{y,G}$ can only partly capture

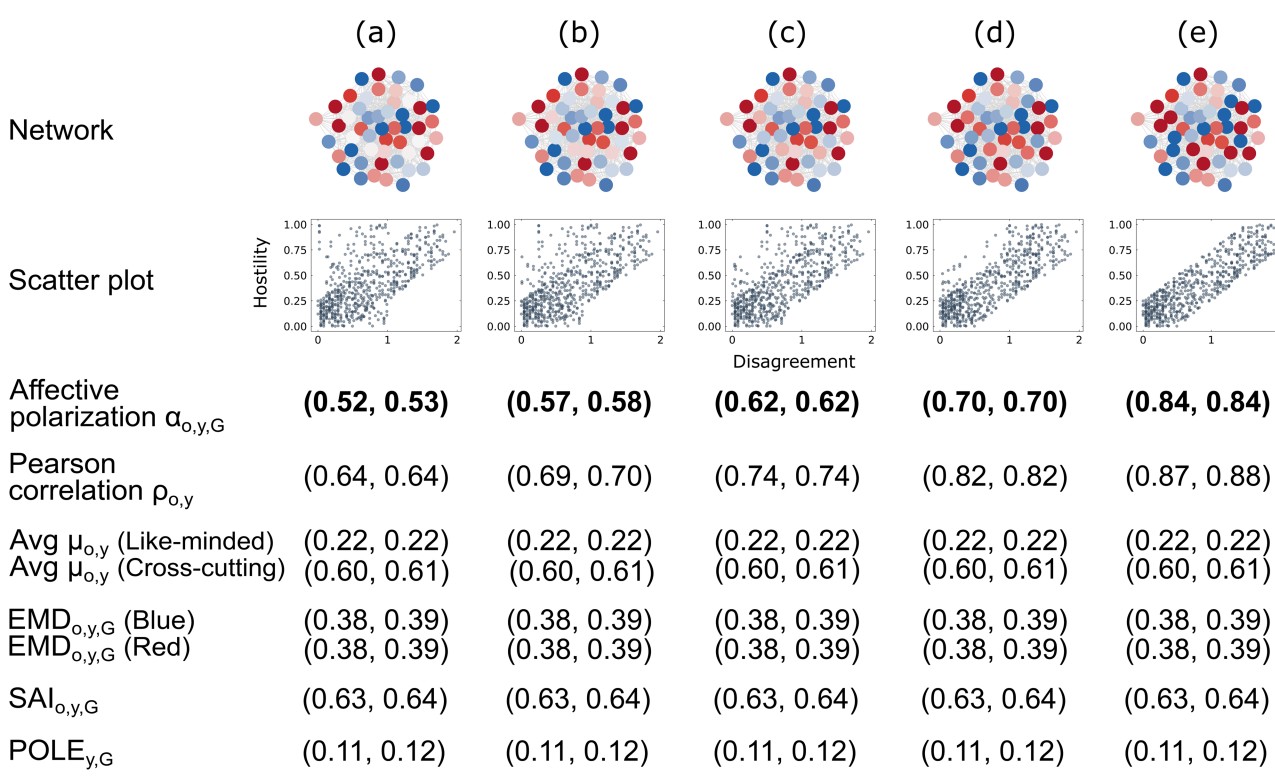

**Fig 3. Affective component (disagreement distribution).** Same legend as Fig 2.

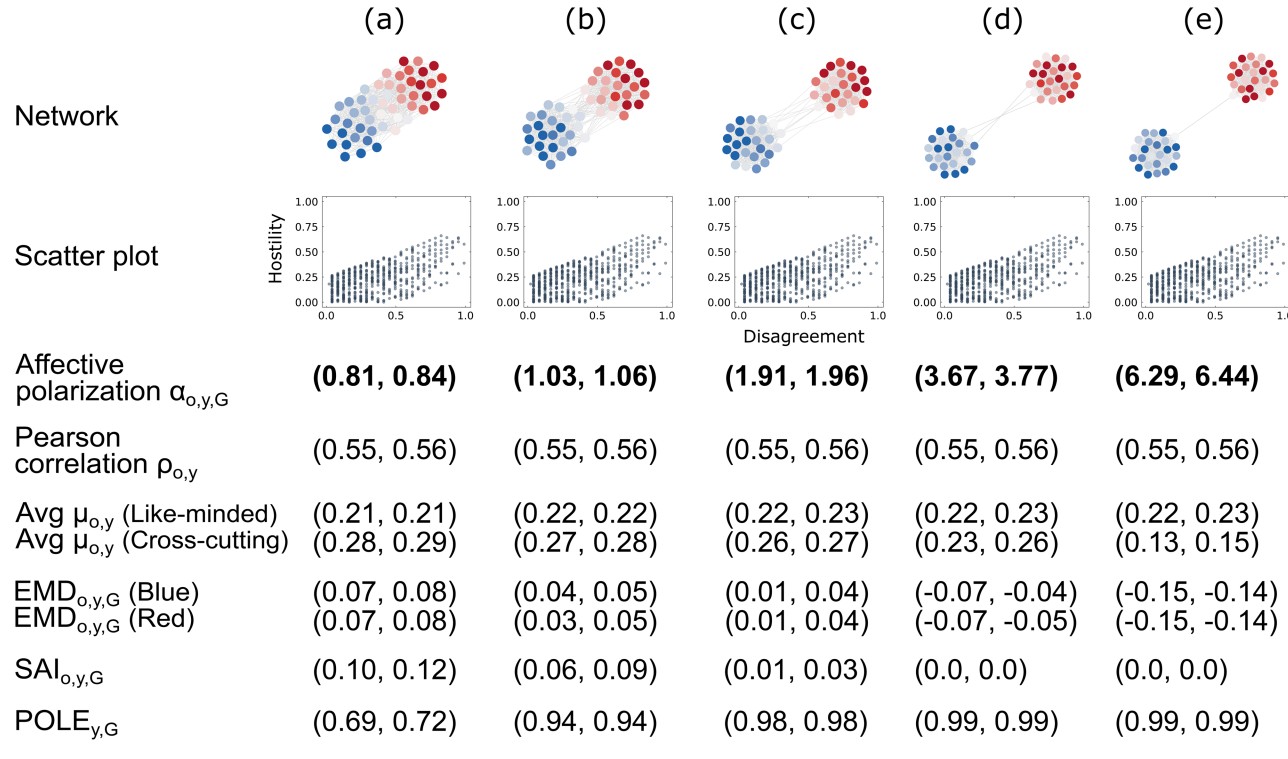

**Fig 4. Social distance component.** Same legend as Fig 2.

this component. While the $POLE_{y,G}$ values increase from (a) to (e), some of the confidence intervals overlap.

$\mu_{o,y}$ *(Cross-cutting)*, $EMD_{o,y,G}$ and $SAI_{o,y,G}$ decrease from (a) to (e), suggesting that network (e) is the least polarized example. This result is not in line with expectations, and these measures consequently do not appropriately capture changes in the social distance component. Lastly, the Pearson correlation coefficient $\rho_{o,y}$ cannot account for changes in this component as the results are the same across all networks.

**Summary.**

Table 1 summarizes the results. From the experiments, we conclude that our proposed measure $\alpha_{o,y,G}$ is the only method tested here that can appropriately capture changes in the hostility and disagreement distributions (affective component) and changes in the structure of social interactions (social distance component). In Sect 2 of the S1 File, we also confirm that $\alpha_{o,y,G}$ can capture changes related to the nodes' opinion strength. In this additional experiment, we show that $\alpha_{o,y,G}$ is the only method that can detect an increase in affective polarization due to uniformly more extreme opinions.

Importantly, our affective polarization measure $\alpha_{o,y,G}$ can summarize the affective and the social distance component in a single affective polarization score. This enables comparisons across several networks, for instance, of people discussing different political issues. For longitudinal analyses, like the Twitter analysis we perform below, it is useful to obtain a single score for each network to track changes over time.

Simultaneously, we can decompose this score to study the two components independently. The Spearman rank correlation coefficient $\tilde{\rho}_{o,y}$ quantifies the affective component, while the Generalized Euclidean distance $\delta_{G,o}$ measures the social distance component. As we show in the following section, analyzing $\tilde{\rho}_{o,y}$ and $\delta_{G,o}$ separately provides further insights into the drivers of affective polarization.

## Twitter data

In addition to the synthetic data presented above, we apply our measure $\alpha_{o,y,G}$ to real-world social media data. Specifically, we analyze a large-scale data set of tweets discussing COVID-19 restrictions between February and July 2020. This allows us to trace the development of affective polarization in the COVID-19 debate on Twitter during the first half year of the pandemic.

By assessing the retweet behavior observed in a sample of 47 million pandemic-related tweets, we infer the ideological leaning of 95,000 Twitter users (see Materials and Methods for details). We use a toxic language classifier to determine how friendly or hostile these users are toward others. Focusing only on tweets containing @-mentions or direct replies, we retrieve a user interaction network for each week between February and July 2020. We analyze both

**Table 1**. Summary of the synthetic experiments.

| | Affective component | | Social distance component |
|---|---|---|---|
| | Hostility | Disagreement | |
| $\alpha_{o,y,G}$ | ✓ | ✓ | ✓ |
| $\rho_{o,y}$ | ✓ | ✓ | ✗ |
| $\mu_{o,y}$ | (✓) | ✗ | ✗ |
| $EMD_{o,y,G}$ | ✓ | ✗ | ✗ |
| $SAI_{o,y,G}$ | (✓) | ✗ | ✗ |
| $POLE_{y,G}$ | ✗ | ✗ | (✓) |

the overall level of affective polarization during this time as well as the affective and social distance component separately.

In early February 2020, we find very low affective polarization in the Twitter data, as indicated by $\alpha_{o,y,G}$ values close to 0 (Fig 5A, week 6). This absence of polarization is expected given that COVID-19 had not been declared a global pandemic and was not the primary topic of public debate yet. Consequently, the engagement on Twitter was limited, involving only a few hundred users in the discussion networks in February 2020.

In week 8, $\tilde{\rho}_{o,y}$ sharply increases, signaling a rise in polarization driven by the affective component. This is followed by an increase in the social distance component, $\delta_{G,o}$, the next week. Interestingly, the two components evolve differently: while $\tilde{\rho}_{o,y}$ (hostility among disagreeing users) decreases again after an initial spike, $\delta_{G,o}$ (social distance) remains high during the spring and summer months. This suggests that, initially, there is intense hostility among users who disagree. Over time, these users start avoiding each other, preferentially engaging with those who share the same opinion. As a result, hostility decreases due to fewer interactions between disagreeing users. This underscores the importance of considering both affect and social distance when analyzing affective polarization dynamics.

For the overall level of affective polarization, we find that $\alpha_{o,y,G}$ reaches a first peak in mid-March when the World Health Organization declared COVID-19 a global pandemic. As a response, the United States government introduced a travel ban, and different US states started issuing stay-at-home orders and implementing statewide shutdowns.

We see a second peak in early April when the Centers for Disease Control and Prevention started recommending face masks as a preventive measure, although officials had discouraged wearing face masks until that point [24]. We analyze user references to restrictions-related terms to examine whether this was reflected in Twitter discussions. Notably, the terms *mask* and *test* dominated the Twitter discussion in weeks 15–17, indicating that Twitter users picked up on the face mask policy change in their discussions during April 2020.

Fig 5 shows the highest $\alpha_{o,y,G}$ values in July 2020. The keyword analysis indicates that, apart from the terms *test* and *mask*, the keyword *school* is among the three most frequently used terms in the sample at this point. This coincides with an intense public debate about whether or not to reopen schools after the summer break [25], and the results presented here indicate that Twitter users engaged in this heated debate online.

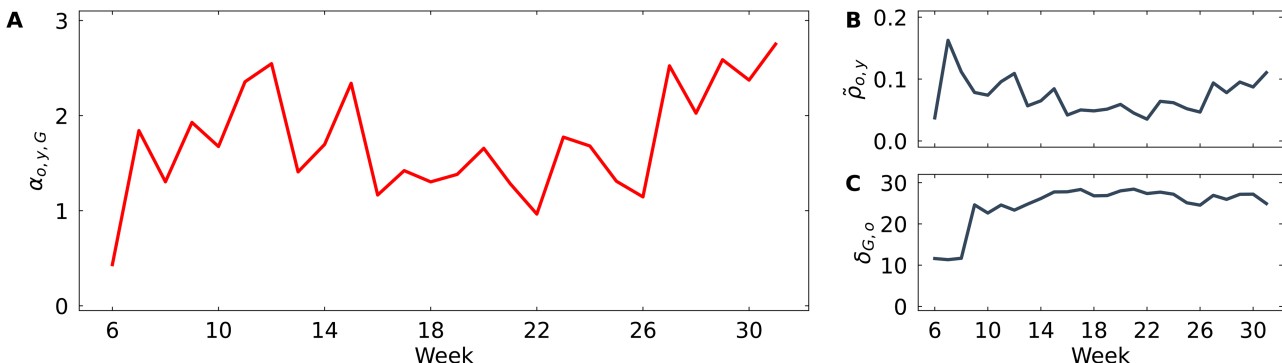

**Fig 5. COVID-19 debate on Twitter.** For each week between February and July 2020, the figure shows: (A) the overall level of affective polarization ($\alpha_{o,y,G}$), (B) the affective component ($\tilde{\rho}_{o,y}$), and (C) the social distance component ($\delta_{G,o}$). (B) and (C) provide a decomposition of the overall affective polarization score shown in (A).

## Discussion

In this paper, we set out to assess affective polarization in a social network across two dimensions: one concerning whether individuals respond to disagreement with hostility (affective component) and the other quantifying the structure of social interactions (social distance component).

We introduce a method that computes the correlation between a disagreement and a hostility vector while considering the structure of the social network. Our measure is sensitive to both components of affective polarization and summarizes them in a single score. As the Twitter data analysis shows, having one unified score proves especially useful for cross-network comparisons and the temporal analysis of affective polarization.

While the ease of comparability is an advantage of this approach, it is important to highlight that the score can also be decomposed into its two components. This facilitates a more in-depth analysis of the factors that drive affective polarization dynamics, as we show in the Twitter analysis. It also allows the use of our method in alternative polarization frameworks, where – differently from here – the social distance component is regarded as a different dimension, rather than a component, of affective polarization.

The work presented here has several limitations that can be addressed in future studies. First, our method relies on a one-dimensional ideological scaling, i.e., it can only consider opinion scores between –1 and +1. When applied to countries other than the United States, we would need a different ideological scale that accounts for a multi-party system. Second, our affective dimension solely focuses on hostility. We do so following the definition predominantly used in the literature. Future work could explore in-group friendliness rather than out-group hostility or analyze other social connections, such as trust, respect, or support.

In addition to introducing an affective polarization measure, we also apply our method to a large-scale Twitter data set in order to analyze the development of online affective polarization during the first half of 2020. The results demonstrate a substantial rise in affective polarization, shifting from low levels in February 2020 to high levels during the spring and summer. These findings align with previous research. We expect elevated levels of polarization in political debates, even on previously unknown topics like COVID-19, given the consistent upward trend in affective polarization in the United States since the 1960s [26]. Our analysis confirms this: as soon as COVID-19 gained public and political attention, along with a critical mass of users engaging in the conversation, affective polarization surged on Twitter.

Further research is needed to clarify in which contexts affective polarization is especially high on social media. This question could, for instance, be tackled by analyzing social media samples collected during election versus non-election times. Since elections are moments of intense partisan conflict, these events will likely be accompanied by increased levels of affective polarization [27,28].

The Twitter data analysis we present is subject to several limitations, especially concerning the sampling strategy. Our data collection relies on a compilation of COVID-19-related tweets from prior research [29]. The initial set of keywords used for sampling did not include terms that reflect criticism, ridicule, or skepticism towards the pandemic. Consequently, the data set primarily comprises liberal-leaning users, while the representation of pandemic-skeptic users is relatively limited (see Sect 3 in the S1 File). While this strong liberal leaning is consistent with other analyses on the ideological divide in the US-based pandemic debate on Twitter [30], the results should nevertheless be interpreted cautiously, given that the sampling might not be representative. Moreover, it is important to note that we only consider a small fraction of users and tweets in the final analysis due to the different preprocessing and filtering steps (see Sect 3 in the S1 File). This problem is common to many studies of polarization

on social media, and further insight is needed on how representative the results are for the entirety of social media users.

## Materials and methods

### Data and code availability

The code needed to replicate the results of the synthetic experiments can be found at https://github.com/marilenahohmann/affective-polarization-measure.

The Twitter data collection and analysis complied with the terms and conditions of the Twitter API at the time of retrieval (spring 2022). Since this data set contains personally identifiable data relating to natural persons and the processing took place within the European Union, it is subject to the General Data Protection Regulation (GDPR). Due to these regulations, we cannot make the Twitter data publicly available.

### Synthetic data generation

This section summarizes how $G$, $o$, and $y$ are generated in the different experiments:

**Network $G$.** We use two different approaches to create the networks: In Figs 2 and 3, we generate a random $G_{n,m}$ graph with $n = 50$ nodes and $m = 610$ edges. The network structure remains the same throughout all examples shown in these figures.

In Fig 4, we generate two cliques of 25 nodes connected by one edge. This network has the same density as the random graphs in Figs 2 and 3. Then, we rewire the network by replacing an edge *within* one of the cliques by an edge *between* the cliques. While rewiring the network, we keep the disagreement value of the old and new edge the same ($\pm$ a margin of 0.05). The number of rewired edges is specified by the parameter $r$: (a) $r = 150$; (b) $r = 50$; (c) $r = 10$; (d) $r = 1$; (e) $r = 0$.

**Opinion vector $o$.** In Figs 2 and 4, we generate 25 equispaced opinion values $o_i \in [0.01, 1]$. The opinion values are summarized in a vector $o^*$ which we use to construct the final vector $o = (o^*, -o^*)$.

In Fig 3, we again start by generating an opinion vector $o^*$ comprising 25 equispaced opinion values $o_i \in [0.01, 1]$. We modify this vector by replacing the first 3 opinions with a specified value: (a) $o_1 = o_2 = o_3 = 0.01$; (b) $o_1 = o_2 = o_3 = 0.125$; (c) $o_1 = o_2 = o_3 = 0.25$; (d) $ao_1 = o_2 = o_3 = 0.5$; (e) $o_1 = o_2 = o_3 = 1.0$. Then, the final opinion vector is $(o^*, -o^*)$.

**Disagreement vector $x$.** Across all experiments, we obtain the disagreement vector $x$ by calculating the absolute opinion difference $x_{i,j} = |o_i - o_j|$ for each pair of nodes directly connected by an edge in $G$. The disagreement values are between 0 (no disagreement) and 2 (maximum disagreement).

**Hostility vector $y$.** The hostility vector $y$ is derived from the disagreement vector $x$. Each entry $y_{i,j}$ is drawn from a random uniform distribution with bounds $x_{i,j}/2 \pm a$, where $a$ determines the strength of the correlation. If $a = 0$, then $y_{i,j} = x_{i,j}/2$, resulting in a perfect linear correlation. If $a = 1$, then $y_{i,j}$ can be any value between 0 and 1, leading $y$ to be uncorrelated with $x$. We can generate negative correlations by taking $1 - (x_{i,j}/2 \pm a)$ instead. In Figs 3 and 4, we generate a positive correlation with $a = 0.25$.

In Fig 2, we use the old hostility value $y_{i,j}$ and a parameter $b$ to modify the distribution. We re-draw each new hostility value $y'_{i,j}$ from a random uniform distribution with bounds $y_{i,j} \pm b$. Importantly, we impose two constraints on these intervals: If $y_{i,j} \leq 0.5$, the interval is limited by 0 as the lower and 0.5 as the upper bound; if $y_{i,j} > 0.5$, it is limited by 0.5 as the lower and 1.0 as the upper bound. In other words, we ensure that any initial hostility values $y_{i,j} \leq 0.5$ remain within $[0,0.5]$, and any values $y_{i,j} > 0.5$ remain within $(0.5,1.0]$. We introduce these

constraints to highlight a shortcoming of $EMD_{o,y,G}$, $SAI_{o,y,G}$ and $POLE_{y,G}$: These measures categorize all non-hostile interactions ($y_{i,j} \leq 0.5$) and all hostile interactions ($y_{i,j} > 0.5$) into two distinct groups without considering the actual distribution of hostility values within these groups.

In Fig 2, we specify the following parameters: (a) negative correlation with $a = 0.01$ (we do not modify the initial hostility vector and $b$, therefore, remains unspecified); (b) negative correlation with $a = 0.01$ and $b = 0.25$; (c) no correlation: $a = 1.0$, $b$ is unspecified; (d) positive correlation with $a = 0.01$, $b = 0.25$; (e) positive correlation with $a = 0.01$, $b$ is unspecified.

## Twitter data collection

Our data collection draws on a large, longitudinal COVID-19 tweet data set [29], previously used in other studies of COVID-19 online discussions [31–36]. This data set contains geo-location information that the authors inferred from the tweet and meta-data available for each tweet. There are five types of location tags available: (1) *geo-coordinates* which are available for users who enabled GPS tracking in their privacy settings; (2) *place-bounding boxes* which contain a GPS location tag within a specific tweet; (3) *profile descriptions* where users can specify a location in a free-text field; (4) *user locations* which public profiles, such as companies, can use to indicate where they are based; and (5) places mentioned in the *tweet text*. Using the geo-locations inferred through (1)–(4) and the tweet IDs provided in this data set, we collect the content, user information, and metadata for tweets by users in the United States between February and July 2020. We focus on US-based users since national borders confined COVID-19 policies, and the online debates surrounding restrictions, therefore, were highly country-specific.

Importantly, our Twitter data set differs from the original one presented in [29] since some users may have deleted or deactivated their accounts. Additionally, the Twitter API only returned publicly accessible tweets, thus excluding messages from users who changed their account privacy settings.

As part of the data preprocessing, we identify all English-language tweets using a pre-trained language detection model [37], and we filter the data set so that the remaining tweets contain at least one keyword related to COVID-19 restrictions. The initial keywords in this list are manually curated and supplemented by semantically similar words, which we found by training a word2vec model [38]. The final data set contains approximately 47 million tweets by 4.1 million users. Sect 3 of the S1 File provides details on the preprocessing and filtering steps.

We rely on users' retweet patterns to determine their political leanings. We compile a list of Twitter accounts for the 118th US Congress members (2019-2021) and assign a score from –1 (liberal) to +1 (conservative) to each account using the politician's DW-NOMINATE scores [39,40]. Additionally, we curate a list of media accounts and we use the political leaning scores provided on mediabiasfactcheck.com to assign a score between –1 and +1 to each news media account. Lastly, we calculate each user's opinion score $o_i$ as the weighted average of the political and media accounts the user retweeted. In the subsequent analysis, we only consider users who retweeted at least five posts to ensure that the opinion scores reflect an actual preference for liberal or conservative content and are not just the result of retweeting viral posts.

Next, we quantify whether users address each other in a hostile manner. While previous studies have often relied on sentiment [4–6,10], we choose toxic language instead [7]. The overall sentiment in the COVID-19 debate was very negative because of the topics users discuss, such as death, disease, or isolation, which should not be confused with out-group

hostility. We, therefore, use a toxicity classifier to determine whether users resort to toxic language when addressing others they disagree with [37].

Lastly, we partition the data set into subsets, each covering a week from Monday to Sunday to mitigate any potential weekday or weekend effects. Within these subsets, we collect all direct interactions – replies and @-mentions – between two individuals, provided we have their respective opinion scores. We use the opinion scores to calculate a disagreement value $x_{i,j}$ for each direct edge between two users. Since there can be more than one reply or @-mention between two users, we calculate the average toxicity of all their interactions and use those as the hostility values $y_{i,j}$. We extract the largest connected component from each undirected interaction network formed by replies and @-mentions. We use the largest connected component and the $o$, $x$, and $y$ vectors to calculate the affective polarization score $\alpha_{o,y,G}$ for each week in the data set.

## Supporting information

**S1 File. Supplementary materials**. Document including supporting analyses, figures, tables, and references.
(PDF)

## Author contributions

**Conceptualization:** Marilena Hohmann, Michele Coscia.

**Data curation:** Marilena Hohmann.

**Formal analysis:** Marilena Hohmann.

**Methodology:** Marilena Hohmann, Michele Coscia.

**Software:** Marilena Hohmann, Michele Coscia.

**Visualization:** Marilena Hohmann.

**Writing – original draft:** Marilena Hohmann, Michele Coscia.

**Writing – review & editing:** Marilena Hohmann, Michele Coscia.

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
