## [Decision Letter · Decision Letter 0]

14 Feb 2025

PONE-D-24-24548Estimating affective polarization on a social networkPLOS ONE

Dear Dr. Hohmann,

Thank you for submitting your manuscript to PLOS ONE. After careful consideration, we feel that it has merit but does not fully meet PLOS ONE’s publication criteria as it currently stands. Therefore, we invite you to submit a revised version of the manuscript that addresses the points raised during the review process.

We look forward to receiving your revised manuscript.

Kind regards,

Carlos Henrique Gomes Ferreira, Ph.D.

Academic Editor

PLOS ONE

Journal Requirements:

2.  In your Methods section, please include additional information about your dataset and ensure that you have included a statement specifying whether the collection and analysis method complied with the terms and conditions for the source of the data.

**Additional Editor Comments:**

Dear Authors,

Unfortunately, the previous Editor of the journal did not complete the editing process for your article.

We apologize for the inconvenience.

I took over yesterday and carefully reviewed the reviewers' comments.

I believe the article is close to being accepted. One reviewer requested a few additional changes, but I believe they are feasible to implement.

Best,

Carlos

Reviewers' comments:

Reviewer's Responses to Questions

**Comments to the Author**

1. Is the manuscript technically sound, and do the data support the conclusions?

Reviewer #1: Yes

Reviewer #2: Partly

2. Has the statistical analysis been performed appropriately and rigorously? 

Reviewer #1: Yes

Reviewer #2: Yes

3. Have the authors made all data underlying the findings in their manuscript fully available?

Reviewer #1: No

Reviewer #2: Yes

4. Is the manuscript presented in an intelligible fashion and written in standard English?

Reviewer #1: Yes

Reviewer #2: Yes

5. Review Comments to the Author

Reviewer #1: Hohmann & Coscia present a nice paper which introduces a novel measure of affective polarization in social networks. The measure is an extension of their excellent work previously published in Science Advances which introduced a structural measure of polarization but with an added consideration for the presence of interaction hostility. The work has a clear motivation, is technically sound, and makes a strong contribution to the interdisciplinary literature on polarization.

In my opinion, considering the target venue is PLOS One, I am of the opinion that the article should be accepted for publication with minimal delay - all requirements of technical correctness and novelty have been met. For completeness, I offer some thoughts and suggestions below. However, I stress these are largely my opinion and should certainly not be seen as a requirement for publication - the authors can feel free to ignore these points if they disagree.

Thoughts and suggestions

1. As the authors well know, how to measure polarization is itself a polarizing topic - different research fields approach the problem in different ways. The authors' measure combines that classic affective component, with the structural dimension. However, I think there are many polarization researchers who might argue that social distance (which your measure captures) is not really needed in a measure of AP, it is simply a common symptom. Some people do include the idea of social distance, but most don't, so IMO, it would be best to try to be more explicit about what you are doing. As part of this, I don't think referring to the "structural" component of AP is particularly clear. Rather, I think the framing in terms of social distance offers more clarity. I would perhaps suggest reframing title / abstract to explicitly highlight this novel contribution of your work, e.g. "A social network measure of affective polarization which captures both hostility and social distance". You could also perhaps touch on why researchers may, or may not, want to include these different elements in their measure.

2. You could be bolder in stating the benefits of your approach and possible useful applications.

3. Related to point 1, I would perhaps suggest not to refer to the survey-based approaches as having shortcomings. There is no consensus as to which ingredients of AP must be included in a useful AP measure, and which are optional. In principle, I think many would argue that social distance is not necesarilly a component of AP, despite being correlated. Rather I think your framing may be stronger by talking about your added value instead of the shortcomings of others. Also, what you call the structural component others may refer to as interactional polarization (see your Ref 11 and Yarchi et al. 2021 in Political Communication).

4. I note you refer to other methods which use sentiment, but that you instead use toxicity. I would politely note that at least one of the papers referenced (Ref 11) is also using toxicity, not sentiment as you state.

5. On page 5 you state: "This opinion homophily indicates high structural separation and, therefore, high affective polarization." I understand this argument, but there is plenty of data to suggest that hostile out-group interactions are not so rare (any of the many papers which argue that echo chambers are in fact rare / non-existant). It is very dependent on interaction type (see for example Fig 4 in Mekacher et al. 2023 PNAS Nexus). The measures of social distance typically used are quick high-stakes interactions (e.g., getting married, being neighbors) unlike social media interactions which are low stakes and are therefore more likely to include out-group interactions.

6. You should perhaps briefly note the assumptions underlying your measure's desire for a positive correlation between disagreement and hostility. If by disagreement you mean disagreement in terms of ideological identification that is fine, but that is not necessarily the same as disagreement on actual issues. See for example the papers by Liliana Mason (I Disrespectfully disagree) which discussed how, surprisingly, affective polarization often manifests as substantial hostility between partisan opponents even when those individuals largely *agree* on the details of policy.

7. In your AP measure, any particular reason you chose to equally weight structure and affect?

8. The method validation section could be a little improved in terms of structure. You are using a lot of acronyms which some readers may not be familiar with. You could take a little more time to introduce the other measures and note why they have been used in the past.

9. Personally, I find Fig 2 a bit messy with all the different metrics included in one figure. I'd probably move most of the other metric results to the SI, but its up to you.

10. Lines 274-275. When you talk about ideological labelling, remember to reference the methods section, otherwise we are confused how you have done this.

11. In Figure 5, perhaps consider showing the individual components of the AP measure and then commenting on how the individual components compare to the combined measure.

12. Lines 336 - 337: "These findings align with previous research indicating a consistent upward trend in affective polarization since the 1960s". Statements like this should be precise about exactly which country you are referring to. I note that AP has decreased in many countries! See Boxell & Gentzkow's NBER paper on AP across countries.

Reviewer #2: The online system keep saying "Minimum Character Count Not Met", even if the character count is 3500. I don't know why. Please find the comments for the authors as a separated file attached to this form.

6. PLOS authors have the option to publish the peer review history of their article (what does this mean?). If published, this will include your full peer review and any attached files.

Reviewer #1: No

Reviewer #2: No

---

## [Author Response · Author response to Decision Letter 1]

31 Mar 2025

We have included a response to the reviewers in a separate file.

---

## [Decision Letter · Decision Letter 1]

16 Jun 2025

PONE-D-24-24548R1Estimating affective polarization on a social networkPLOS ONE

Dear Dr. Hohmann,

Thank you for submitting your manuscript to PLOS ONE. After careful consideration, we feel that it has merit but does not fully meet PLOS ONE’s publication criteria as it currently stands. Therefore, we invite you to submit a revised version of the manuscript that addresses the points raised during the review process.

In particular, I would like to thank you for the opportunity to continue the review process of this manuscript. We would like to note that two of the reviewers now consider the article ready for publication.

However, I third reviewer, while acknowledging the authors’ efforts and the lack of consensus in the field, raises an point regarding the conceptual distinction between affective and structural dimensions of polarization.

Although this reviewer ultimately respects the authors’ position, their comments suggest that, at a minimum, the manuscript should explicitly acknowledge this distinction and critically engage with the differing perspectives in the literature.

We kindly ask the authors to consider this in their revision, ensuring that the current framing is clearly justified and that alternative interpretations are adequately addressed.

We would like to note that two of the reviewers now consider the article ready for publication.

However, a third one, while acknowledging the authors’ efforts and the lack of consensus in the field, raises an point regarding the conceptual distinction between affective and structural dimensions of polarization.

Although this reviewer ultimately respects the authors’ position, their comments suggest that, at a minimum, the manuscript should explicitly acknowledge this distinction and critically engage with the differing perspectives in the literature.

Thus, we kindly ask the authors to consider this in their revision.

We look forward to receiving your revised manuscript.

Kind regards,

Carlos Henrique Gomes Ferreira, Ph.D.

Academic Editor

PLOS ONE

Journal Requirements:

Reviewers' comments:

Reviewer's Responses to Questions

**Comments to the Author**

1. If the authors have adequately addressed your comments raised in a previous round of review and you feel that this manuscript is now acceptable for publication, you may indicate that here to bypass the “Comments to the Author” section, enter your conflict of interest statement in the “Confidential to Editor” section, and submit your "Accept" recommendation.

Reviewer #1: All comments have been addressed

Reviewer #2: (No Response)

Reviewer #3: All comments have been addressed

2. Is the manuscript technically sound, and do the data support the conclusions?

Reviewer #1: Yes

Reviewer #2: Partly

Reviewer #3: Yes

3. Has the statistical analysis been performed appropriately and rigorously? 

Reviewer #1: Yes

Reviewer #2: Yes

Reviewer #3: Yes

4. Have the authors made all data underlying the findings in their manuscript fully available?

Reviewer #1: Yes

Reviewer #2: Yes

Reviewer #3: Yes

5. Is the manuscript presented in an intelligible fashion and written in standard English?

Reviewer #1: Yes

Reviewer #2: Yes

Reviewer #3: Yes

6. Review Comments to the Author

Reviewer #1: The authors have done an excellent job of responding to all comments. I recommend publication with no further revisions.

Reviewer #2: First, I would like to thank the authors for their effort in addressing my previous concerns. However, I still have concerns regarding the structural aspects analyzed in this work, now referred to as "social distance". While some studies consider this aspect a component or manifestation of affective polarization, others treat it as a distinct dimension of polarization. Personally, I align with the latter perspective and believe that affective and structural dimensions should not be conflated when measuring affective polarization, particularly in the context of social media.

Indeed, social media platforms rely heavily on recommendation algorithms, which significantly shape the structure of interactions by promoting content aligned with users’ interests. In this context, the structural component may represent a mix of social avoidance (which I see as an effect, not a cause, of affective polarization) and algorithmic design.

Even the results presented in Figure 5 could support this interpretation: the affective component appears to precede the structural one. Initially, there are more interactions across communities; however, due to heightened affective polarization, users begin to avoid cross-community engagement, which in turn increases the structural component, or “social distance.”

In my view, the paper would benefit from being reframed as a comparison between these two dimensions of polarization. That said, I respect the authors’ position on the matter and recognize that this remains an open question without a clear consensus in the field.

Reviewer #3: The paper has merit for publication in PLOS ONE. The proposed measure contributes to the research area as an useful method to analyse the factors that drive affective polarization dynamics in online social media platforms.

7. PLOS authors have the option to publish the peer review history of their article (what does this mean?). If published, this will include your full peer review and any attached files.

Reviewer #1: No

Reviewer #2: No

Reviewer #3: No

---

## [Author Response · Author response to Decision Letter 2]

20 Jun 2025

Reviewer #2:

I still have concerns regarding the structural aspects analyzed in this work, now referred to as "social distance''. While some studies consider this aspect a component or manifestation of affective polarization, others treat it as a distinct dimension of polarization. [...] In my view, the paper would benefit from being reframed as a comparison between these two dimensions of polarization. That said, I respect the authors’ position on the matter and recognize that this remains an open question without a clear consensus in the field.

Authors' Answer:

We think the reviewer has a solid point here. We have edited abstract, introduction, and discussion to improve our acknowledgment of this open question -- and to allow the reader siding with the reviewer to still find usefulness in the method we develop by considering exclusively the affective component and relegating the social distance component as a distinct dimension.

---

## [Editor Report · Decision Letter 2]

30 Jun 2025

Estimating affective polarization on a social network

PONE-D-24-24548R2

Dear Dr. Hohmann,

We’re pleased to inform you that your manuscript has been judged scientifically suitable for publication and will be formally accepted for publication once it meets all outstanding technical requirements.

Kind regards,

Carlos Henrique Gomes Ferreira, Ph.D.

Academic Editor

PLOS ONE

Additional Editor Comments (optional):

I observed that the authors have addressed the point raised by the reviewer. With that, I consider the article ready for publication.

Congratulations on the work!
---

## [Editor Report · Acceptance letter]

PONE-D-24-24548R2

PLOS ONE

Dear Dr. Coscia,

I'm pleased to inform you that your manuscript has been deemed suitable for publication in PLOS ONE. Congratulations! Your manuscript is now being handed over to our production team.

Kind regards,

on behalf of

Dr. Carlos Henrique Gomes Ferreira

Academic Editor

PLOS ONE